# Fast amortized inference of neural activity from calcium imaging data with variational autoencoders

**Artur Speiser**[12]**, Jinyao Yan**[3]**, Evan Archer**[4][*]**, Lars Buesing**[4][†]**,**
**Srinivas C. Turaga**[3][‡] **and Jakob H. Macke**[1][†][§]
[1]research center caesar, an associate of the Max Planck Society, Bonn, Germany
[2]IMPRS Brain and Behavior Bonn/Florida
[3]HHMI Janelia Research Campus
[4]Columbia University
artur.speiser@caesar.de, turagas@janelia.hhmi.org, jakob.macke@caesar.de

## Abstract

Calcium imaging permits optical measurement of neural activity. Since intracellular calcium concentration is an indirect measurement of neural activity, computational tools are necessary to infer the true underlying spiking activity from fluorescence measurements. Bayesian model inversion can be used to solve this problem, but typically requires either computationally expensive MCMC sampling, or faster but approximate maximum-a-posteriori optimization. Here, we introduce a flexible algorithmic framework for fast, efficient and accurate extraction of neural spikes from imaging data. Using the framework of variational autoencoders, we propose to amortize inference by training a deep neural network to perform model inversion efficiently. The recognition network is trained to produce samples from the posterior distribution over spike trains. Once trained, performing inference amounts to a fast single forward pass through the network, without the need for iterative optimization or sampling. We show that amortization can be applied flexibly to a wide range of nonlinear generative models and significantly improves upon the state of the art in computation time, while achieving competitive accuracy. Our framework is also able to represent posterior distributions over spike-trains. We demonstrate the generality of our method by proposing the first probabilistic approach for separating backpropagating action potentials from putative synaptic inputs in calcium imaging of dendritic spines.

## 1 Introduction

Spiking activity in neurons leads to changes in intra-cellular calcium concentration which can be measured by fluorescence microscopy of synthetic calcium indicators such as Oregon Green BAPTA-1 [1] or genetically encoded calcium indictors such as GCaMP6 [2]. Such calcium imaging has become important since it enables the parallel measurement of large neural populations in a spatially resolved and minimally invasive manner [3, 4]. Calcium imaging can also be used to study neural activity at subcellular resolution, e.g. for measuring the tuning of dendritic spines [5, 6]. However, due to the indirect nature of calcium imaging, spike inference algorithms must be used to infer the underlying neural spiking activity leading to measured fluorescence dynamics.

---

[*]current affiliation: Cogitai.Inc
[†]current affiliation: DeepMind
[‡]equal contribution
[§]current primary affiliation: Centre for Cognitive Science, Technical University Darmstadt

Most commonly-used approaches to spike inference [7, 8, 9, 10, 11, 12, 13, 14] are based on carefully designed generative models that describe the process by which spiking activity leads to fluorescence measurements. Spikes are treated as latent variables, and spike-prediction is performed by inferring both the parameters of the model and the spike latent variables from fluorescence time series, or "traces" [7, 8, 9, 10]. The advantage of this approach is that it does not require extensive ground truth data for training, since simultaneous electrophysiological and fluorescence recordings of neural activity are difficult to acquire, and that prior knowledge can be incorporated in the specification of the generative model. The accuracy of the predictions depends on the faithfulness of the generative model of the transformation of spike trains into fluorescence measurements [14, 12]. The disadvantage of this approach is that spike-inference requires either Markov-Chain Monte Carlo (MCMC) or Sequential Monte-Carlo techniques to sample from the posterior distribution over spike-trains or alternatively, iterative optimization to obtain an approximate maximum a-posteriori (MAP) prediction. Currently used approaches rely on bespoke, model-specific inference algorithms, which can limit the flexibility in designing suitable generative models. Most commonly used methods are based on simple phenomenological (and often linear) models [7, 8, 9, 10, 13].

Recently, a small number of cell-attached electrophysiological recordings of neural activity have become available, with simultaneous fluorescence calcium measurements in the same neurons. This has made it possible to train powerful and fast classifiers to perform spike-inference in a discriminative manner, precluding the need for accurate generative models of calcium dynamics [15]. The disadvantage of this approach is that it can require large labeled data-sets for every new combination of calcium indicator, cell-type and microscopy method, which can be expensive or impossible to acquire. Further, these discriminative methods do not easily allow the incorporation of prior knowledge about the generative process. Finally, current classification approaches yield only pointwise predictions of spike probability (i.e. firing rates), independent across time, and ignore temporal correlations in the posterior distribution of spikes.

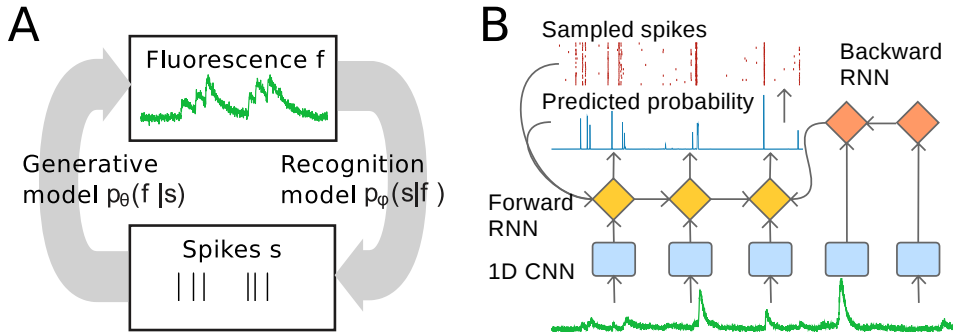

Figure 1: **Amortized inference for predicting spikes from imaging data. A)** Our goal is to infer a spike train $s$ from an observed time-series of fluorescence-measurements $f$. We assume that we have a generative model of fluorescence given spikes with (unknown) parameters $\theta$, and we simultaneously learn $\theta$ as well as a 'recognition model' which approximates the posterior over spikes $s$ given $f$ and which can be used for decoding a spike train from imaging data. **B)** We parameterize the recognition-model by a multi-layer network architecture: Fluorescence-data is first filtered by a deep 1D convolutional network (CNN), providing input to a stochastic forward running recurrent neural network (RNN) which predicts spike-probabilities and takes previously sampled spikes as additional input. An additional deterministic RNN runs backward in time and provides further context.

Here, we develop a new spike inference framework called DeepSpike (DS) based on the variational autoencoder technique which uses stochastic variational inference (SVI) to teach a classifier to predict spikes in an unsupervised manner using a generative model. This new strategy allows us to combine the advantages of generative [7] and discriminative approaches [15] into a single fast classifier-based method for spike inference. In the variational autoencoder framework, the classifier is called a *recognition model* and represents an approximate posterior distribution over spike trains from which samples can be drawn in an efficient manner. Once trained to perform spike inference on one dataset, the recognition model can be applied to perform inference on statistically similar datasets without any retraining: The computational cost of variational spike inference is *amortized*, dramatically speeding up inference at test-time by exploiting fast, classifier based recognition models.

We introduce two recognition models: The first is a temporal convolutional network which produces a posterior distribution which is factorized in time, similar to standard classifier-based methods [15]. The second is a recurrent neural network-based recognition model, similar to [16, 17] which can represent any correlated posterior distribution in the non-parametric limit. Once trained, both models perform spike inference with state-of-the-art accuracy, and enable simultaneous spike inference for populations as large as $10^4$ in real time on a single GPU.

We show the generality of this black-box amortized inference method by demonstrating its accuracy for inference with a classic linear generative model [7, 8], as well as two nonlinear generative models [12]. Finally, we show an extension of the spike inference method to simultaneous inference and demixing of synaptic inputs from backpropagating somatic action potentials from simultaneous somatic and dendritic calcium imaging.

## 2 Amortized inference using variational autoencoders

### 2.1 Approach and training procedure

We observe fluorescence traces $f_t^i$, $t = 1 \ldots T^i$ representing noisy measurements of the dynamics of somatic calcium concentration in neurons $i = 1 \ldots N$. We assume a parametrised, probabilistic, differentiable generative model $p_{\theta^i}(f|s)$ with (unknown) parameters $\theta^i$. The generative model predicts a fluorescence trace given an underlying binary spike train $s^i$, where $s_t^i = 1$ indicates that the neuron $i$ produced an action potential in the interval indexed by $t$. Our goal is to infer a latent spike-train $s$ given only fluorescence observations $f$. We will solve this problem by training a deep neural network as a "recognition model" [18, 19, 20] parametrized by weights $\phi$. Use of a recognition model enables fast computation of an approximate posterior distribution over spike trains from a fluorescence trace $q_\phi(s|f)$. We will share one recognition model across multiple cells, i.e. that $q_\phi(s^i|f^i) \approx p_{\theta^i}(s^i|f^i)$ for each $i$. We describe an unsupervised training procedure which jointly optimizes parameters of the generative model $\theta$ and the recognition network $\phi$ in order to maximize a lower bound on the log likelihood of the observed data, $\log p(f)$ [19, 18, 20].

We learn the parameters $\phi$ and $\theta$ simultaneously by jointly maximizing $\mathcal{L}^K(\theta, \phi)$, a multi-sample importance-weighting lower bound on the log likelihood $\log p(f)$ given by [21]

$$\mathcal{L}^K(\theta, \phi) = \mathbb{E}_{s^1,\ldots,s^K \sim q_\phi(s|f)} \left[ \log \frac{1}{K} \sum_{k=1}^{K} \frac{p_\theta(s^k, f)}{q_\phi(s^k|f)} \right] \leq \log p(f), \tag{1}$$

where $s^k$ are spike trains sampled from the recognition model $q_\phi(s|f)$. This stochastic objective involves drawing $K$ samples from the recognition model, and evaluating their likelihood by passing them through the generative model. When $K = 1$, the bound reduces to the *evidence lower bound* (ELBO). Increasing $K$ yields a tighter lower bound (than the ELBO) on the marginal log likelihood, at the cost of additional training time. We found that increasing the number of samples leads to better fits of the generative model; in our experiments, we used $K = 64$.

To train $\theta$ and $\phi$ by stochastic gradient ascent, we must estimate the gradient $\nabla_{\phi,\theta} \mathcal{L}(\theta, \phi)$. As our recognition model produces an approximate posterior over binary spike trains, the gradients have to be estimated based on samples. Obtaining functional estimates of the gradients $\nabla_\phi \mathcal{L}(\theta, \phi)$ with respect to parameters of the recognition model is challenging and relies on constructing effective control variates to reduce variance [22]. We use the *variational inference for monte carlo objectives* (VIMCO) approach of [23] to produce low-variance unbiased estimates of the gradients $\nabla_{\phi,\theta} \mathcal{L}^K(\theta, \phi)$. The generative training procedure could be augmented with a supervised cost term [24, 25], resulting in a semi-supervised training method.

**Gradient optimization:** We use ADAM [26], an adaptive gradient update scheme, to perform online stochastic gradient ascent. The training data is cut into short chunks of several hundred time-steps and arranged in batches containing samples from a single cell. As we train only one recognition model but multiple generative models in parallel, we load the respective generative model and ADAM parameters at each iteration. Finally, we use norm-clipping to scale the gradients acting on the recognition model: the norm of all gradients is calculated, and if it exceeds a fixed threshold the gradients are rescaled. While norm-clipping was introduced to prevent exploding gradients in RNNs

[27], we found it to be critical to achieve high performance both for RNN and CNN architectures in our learning problem. Very small threshold values (0.02) empirically yielded best results.

## 2.2 Generative models $p_\theta(f|s)$

To demonstrate that our computational strategy can be applied to a wide range of differentiable models in a black-box manner, we consider four generative models: a simple, but commonly used linear model of calcium dynamics [7, 8, 9, 10], two more sophisticated nonlinear models which additionally incorporate saturation and facilitation resulting from the dynamics of calcium binding to the calcium sensor, and finally a multi-dimensional model for dendritic imaging data.

**Linear auto-regressive generative model (SCF):** We use the name *SCF* for the classic linear convolutional generative model used in [7, 8, 9, 10], since this generative process is described by the Spikes $s_t$, which linearly impact Calcium concentration $c_t$, which in turn determines the observed Fluorescence intensity $f_t$,

$$c_t = \sum_{t'=1}^{p} \gamma_{t'} c_{t-t'} + \delta s_t, \qquad f_t = \alpha c_t + \beta + e_t, \tag{2}$$

with linear auto-regressive dynamics of order $p$ for the calcium concentration with parameters $\gamma$, spike-amplitude $\delta$, gain $\alpha$, constant fluorescence baseline $\beta$, and additive measurement noise $e_t \sim \mathcal{N}(0, \sigma^2)$.

**Nonlinear auto-regressive and sensor dynamics generative models (SCDF & MLphys):** As examples of nonlinear generative models [28], we consider two simple models of the discrete-time dynamics of the calcium sensor or dye. In the first (SCDF), the concentration of fluorescent dye molecules $d_t$ is a function of the somatic Calcium concentration $c_t$, and has Dynamics

$$d_t - d_{t-1} = \kappa_{\text{on}} c_t^\eta ([D] - d_{t-1}) - \kappa_{\text{off}} d_{t-1}, \qquad f_t = \alpha d_t + \beta + e_t, \tag{3}$$

where $\kappa_{\text{on}}$ and $\kappa_{\text{off}}$ are the rates at which the calcium sensor binds and unbinds calcium ions, and $\eta$ is a Hill coefficient. We constrained these parameters to be non-negative. $[D]$ is the total concentration of the dye molecule in the soma, which sets the maximum possible value of $d_t$. The richer dynamics of the SCDF model allow for facilitation of fluorescence at low firing rates, and saturation at high rates. The parameters of the SCDF model are $\theta = \{\alpha, \beta, \gamma, \kappa_{\text{on}}, \kappa_{\text{off}}, \eta, [D], \sigma^2\}$.

The second nonlinear model (MLphys) is a discrete-time version of the MLspike generative model [12], simplified by not including a model of the time-varying baseline. The dynamics for $f_t$ and $c_t$ are as above, with $\delta = 1$. We replace the dynamics for $d_t$ by

$$d_t - d_{t-1} = \frac{1}{\tau_{on}} (1 + \omega((c_0 + c_t)^\eta - c_0^\eta)) \left( \frac{((c_0 + c_t)^\eta - c_0^\eta)}{(1 + \omega((c_0 + c_t)^\eta - c_0^\eta))} - d_{t-1} \right). \tag{4}$$

**Multi-dimensional soma + dendrite generative model (DS-F-DEN):** The dendritic generative model is a multi-dimensional SCDF model that incorporates back-propagating action potentials (bAPs). The calcium concentration at the cell body (superscript c) is generated as for SCDF, whereas for the spine (superscript s), there are two components: synaptic inputs and bAPs from the soma,

$$c_t^c = \sum_{t'=1}^{p} \gamma_{t'}^c c_{t-t'}^c + \delta^c s_t^c, \qquad c_t^s = \sum_{t'=1}^{p} \gamma_{t'}^s c_{t-t'}^s + \delta^s s_t^s + \delta^{b_s} s_t^c, \tag{5}$$

where $\delta^{b_s}$ are the amplitude coefficients of bAPs for different spine locations, and $c \in \{1, ..., N_c\}$, $s \in \{1, ..., N_s\}$. The spines and soma share the same dye dynamics as in (3). The parameters of the dendritic integration model are $\theta = \{\alpha_{s,c}, \beta_{s,c}, \gamma_{s,c}, \kappa_{\text{on}}, \kappa_{\text{off}}, \eta, [D], \sigma_{s,c}^2\}$. We note that this simple generative model does not attempt to capture the full complexity of nonlinear processing in dendrites (e.g. it does not incorporate nonlinear phenomena such as dendritic plateau potentials). Its goal is to separate local influences (synaptic inputs) from global events (bAPs, or potentially regenerative dendritic events).

## 2.3 Recognition models: parametrization of the approximate posterior $q_\phi(s|f)$

The goal of the recognition model is to provide a fast and efficient approximation $q_\phi(s|f)$ to the true posterior $p(s|f)$ over discrete latent spike trains $s$. We will use both a factorized, localized approximation (parameterized as a convolutional neural network), and a more flexible, non-factorized and non-localized approximation (parameterized using additional recurrent neural networks).

**Convolutional neural network: Factorized posterior approximation (DS-F)**   In [15], it was reported that good spike-prediction performance can be achieved by making the spike probability $q_\phi(s_t|f_{t-\tau...t+\tau})$ depend on a local window of the fluorescence trace of length $2\tau + 1$ centered at $t$ when training such a model fully supervised. We implement a scaled up version of this idea, using a deep neural network which is convolutional in time as the recognition model. We use architectures with up to five hidden layers and $\approx 20$ filters per layer with Leaky ReLUs units [29]. The output layer uses a sigmoid nonlinearity to compute the Bernoulli spike probabilities $q_\phi(s_t|f)$.

**Recurrent neural network: Capturing temporal correlations in the posterior (DS-NF)**   The fully-factorized posterior approximation (DS-F) above ignores temporal correlations in the posterior over spike trains. Such correlations can be useful in modeling uncertainty in the precise timing of a spike, which induces negative correlations between nearby time bins. To model temporal correlations, we developed a RNN-based non-factorizing distribution which can approach the true posterior in the non-parametric limit (see figure 1B). Similar to [16], we use the temporal ordering over spikes and factorize the joint distribution over spikes as $q_\phi(s|f) = \prod_t q_\phi(s_t|f, s_0, ..., s_{t-1})$, by conditioning spikes at $t$ on all previously sampled spikes. Our RNN uses a CNN as described above to extract features from the input trace. Additional input is provided by a a backwards RNN which also receives input from the CNN features. The outputs of the forward RNN and CNN are transformed into Bernoulli spike probabilities $q_\phi(s_t|f)$ through a dense sigmoid layer. This probability and the sample drawn from it are relayed to the forward RNN in the next time step. Forward and backward RNN have a single layer with 64 gated recurrent units each [30].

## 2.4 Details of synthetic and real data and evaluation methodology

We evaluated our method on simulated and experimental data. From our SCF and SCDF generative models for spike-inference, we simulated traces of length $T = 10^4$ assuming a recording frequency of 60 Hz. Initial parameters where obtained by fitting the models to real data (see below), and heterogeneity across neurons was achieved by randomly perturbing parameters. We used 50 neurons each for training and validation and 100 neurons in the test set. For each cell, we generated three traces with firing rates of 0.6, 0.9 and 1.1 Hz, assuming i.i.d. spikes.

Finally, we compared methods on two-photon imaging data from $9 + 11$ cells from [2], which is available at www.crcns.org. Layer 2/3 pyramidal neurons in mouse visual cortex were imaged at 60 Hz using the genetically encoded calcium-indicators GCaMP6s and GCaMP6f, while action-potentials were measured electrophysiologically using cell-attached recordings. Data was pre-processed by removing a slow moving baseline using the 5th percentile in a window of 6000 time steps. Furthermore we used this baseline estimate to calculate $\Delta F/F$. Cross-validated results where obtained using 4 folds, where we trained and validated on 3/4 of the cells in each dataset and tested on the remaining cells to highlight the potential for amortized inference. Early stopping was performed based on the the correlation achieved on the train/validation set, which was evaluated every 100 update steps.

We report results using the cross-correlation between true and predicted spike-rates, at the sampling discretization of 16.6 ms for simulated data and 40 ms for real data. As the predictions of our DS-NF model are not deterministic, we sample 30 times from the model and average over the resulting probability distributions to obtain an estimate of the marginal probability before we calculate cross-correlations.

We used multiple generative models to show that our inference algorithm is not tied to a particular model: SCDF for the experiments depicted in Fig. 2, SCF for a comparison with established methods based on this linear model (Table 1, column 1), and MLphys on real data as it is used by the current state-of-the-art inference algorithm (Table 1, columns 2 & 3, Fig. 3).

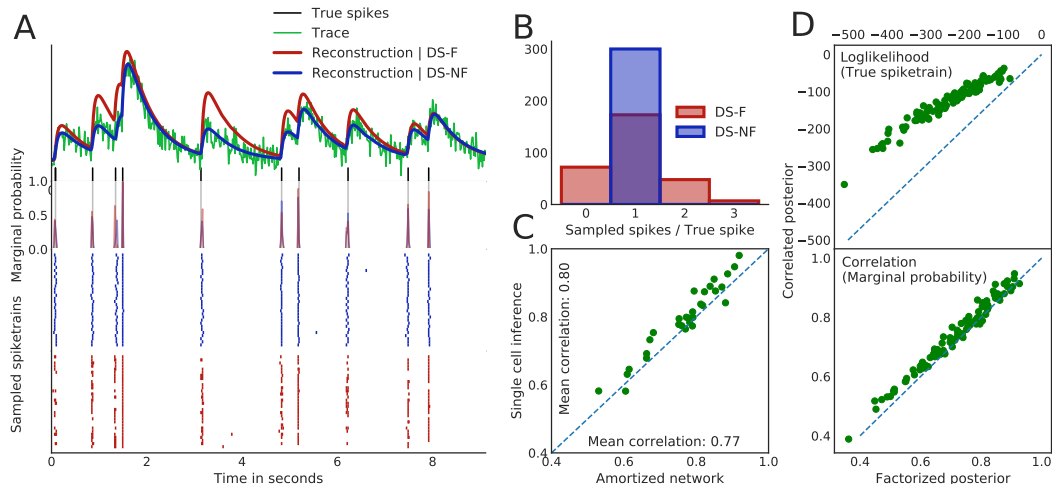

Figure 2: **Model-inversion with variational autoencoders, simulated data A)** Illustration of factorized (CNN, DS-F) and non-factorized posterior approximation (RNN, DS-NF) on simulated data (SCDF generative model). DS-NF yields more accurate reconstructions, but both methods lead to similar marginal predictions (i.e. predicted firing rates, bottom). **B)** Number of spikes sampled for every true spike for the factorized (red) and non-factorized (red) posterior. The correlated posterior consistently samples the correct number of spikes while still accounting for the uncertainty in the spike timing. **C)** Performance of amortized vs non-amortized inference on simulated data. **D)** Scatter plots of achieved log-likelihood of the true spike train under the posterior model (top) and achieved correlation coefficients between the marginalized spiking probabilities and true spike trains (bottom).

# 3 Results

## 3.1 Stochastic variational spike inference of factorized and correlated posteriors

We first illustrate our approach on synthetic data, and compare our two different architectures for recognition models. We simulated data from the SCDF nonlinear generative model and trained DeepSpike unsupervised using the same SCDF model. While only the more expressive recognition model (DS-NF) is able to achieve a close-to-perfect reconstructions of the fluorescence traces (Fig. 2 A, top row), both approaches yield similar marginal firing rate predictions (second row). However, as the factorized model does not model correlations in the posterior, it yields higher variance in the number of spikes reconstructed for each true spike (Fig. 2 B). This is because the factorized model can not capture that a fluorescence increase might be 'explained away' by a spike that has just been sampled, i.e. it can not capture the difference between uncertainty in spike-timing and uncertainty in (local) spike-counts. Therefore, while both approaches predict firing rates similarly well on simulated data (as quantified using correlation, Fig. 2 D), the DS-NF model assigns higher posterior probability to the true spike trains.

## 3.2 Amortizing inference leads to fast and accurate test-time inference

In principle, our unsupervised learning procedure could be re-trained on every data-set of interest. However, it also allows for amortizing inference by sharing one recognition model across multiple cells, and applying the recognition model directly on new data without additional training for fast test-time performance. Amortized inference allows for the recognition model to be used for inference in the same way as a network that was trained fully supervised. Since there is no variational optimization at test time, inference with this network is just as fast as inference with a supervised network. Similarly to supervised learning, there will be limitations on the ability of this network to generalize to different imaging conditions or indicators that where not included in the training set.

To test if our recognition model generalizes well enough for amortized inference to work across multiple cells, as well as on cells it did not see during training, we trained one DS-NF model on 50

cells (simulated data, SCDF) and evaluated its performance on a non-overlapping set of 30 cells. For comparison, we also trained 30 DS-NF models separately, on each of those cells– this amounts to standard variational inference using a neural network to parametrize the posterior approximation, but without amortizing inference. We found that amortizing inference only causes a small drop in performance (Fig. 2 C). However, this drop in performance is offset by the the large gain in computational efficiency as training a neural network takes several orders of magnitude more time then applying it at test time.

Inference using the DS-F model only requires a single forward pass through a convolutional network to predict firing rates, and DS-NF requires running a stochastic RNN for each sampled spike train. While the exact running-time of each of these applications will depend on both implementation and hardware, we give rough indications of computational speed number estimated on an Intel(R) Xeon(R) CPU E5-2697 v3. On the CPU, our DS-F approach takes $0.05\,$s to process a single trace of 10K time steps, when using a network appropriate for $60\,$Hz data. This is on the same order as the $0.07\,$s (Intel Core i5 2.7 GHz CPU) reported by [31] for their OASIS algorithm, which is currently the fastest available implementation for constrained deconvolution (CDEC) of SCF, but restricted to this linear generative model. The DS-NF algorithm requires $4.6\,$s which still compares favourably to MLspike which takes $9.2\,$s (evaluated on the same CPU). As our algorithm is implemented in Theano [32] it can be easily accelerated and allows for massive parallelization on a single GPU. On a GTX Titan X, DS-F and DS-NF take $0.001\,$s and $1.5\,$s, respectively. When processing 500 traces in parallel, DS-NF becomes only 2.5 times slower. Extrapolating from these results, this implies that even when using the DS-NF algorithm, we would be able to perform spike-inference on 1 hour of recordings at $60\,$Hz for 500 cells in less then $90\,$s.

Table 1: Performance comparison. Values are correlations between predicted marginal probabilities and ground truth spikes.

| Algorithm | Dataset | | | Dendritic dataset | |
|---|---|---|---|---|---|
| | SCF-Sim. | GCaMP6s | GCaMP6f | Soma | Spine |
| DS-F | $0.88 \pm 0.01$ | $0.74 \pm 0.02$ | $0.74 \pm 0.02$ | | |
| DS-NF | $0.89 \pm 0.01$ | $0.72 \pm 0.02$ | $0.73 \pm 0.02$ | | |
| CDEC [10] | $0.86 \pm 0.01$ | $0.39 \pm 0.03$ * | $0.58 \pm 0.02$ * | | |
| MCMC [9] | $0.87 \pm 0.01$ | $0.47 \pm 0.03$ * | $0.53 \pm 0.03$ * | | |
| MLSpike [12] | | $0.60 \pm 0.02$ * | $0.67 \pm 0.01$ * | | |
| DS-F-DEN | | | | $0.84 \pm 0.01$ | $0.78 \pm 0.01$ |
| Foopsi-RR [2] | | | | $0.66 \pm 0.02$ | $0.60 \pm 0.01$ |

### 3.3 DS achieves competitive results on simulated and publicly available imaging data

The advantages of our framework (black-box inference for different generative models, fast test-time performance through amortization, correlated posteriors through RNNs) are only useful if the approach can also achieve competitive performance. To demonstrate that this is the case, we compare our approach to alternative generative-model based spike prediction methods on data sampled from the SCF model– as this is the generative model underlying commonly used methods [10, 9], it is difficult to beat their performance on this data. We find that both DS-F and DS-NF achieve competitive performance, as measured by correlation between predicted firing rates and true (simulated) spike trains (Table 1, left column. Values are means and standard error of the mean calculated over cells).

To evaluate our performance on real data we compare to the current state-of-the-art method for spike inference based on generative models[12]. For these experiments we trained separate models on each of the GCaMP variants using the MLspike generative model. We achieve competitive accuracy to the results in [12] (see Table 1, values marked with an asterisk are taken from [12], Fig. 6d) and clearly outperform methods that are based on the linear SCF model. We note that, while our method performs inference in an unsupervised fashion and is trained using an un-supervised objective, we initialized our generative model with the mean values given in [12] (Fig. S6a), which were obtained using ground truth data. An example of inference and reconstruction using the DS-NF model is shown in Fig. 3. The reconstruction based on the true spikes (purple line) was obtained using the generative model parameters which had been acquired from unsupervised learning. This explains why the reconstruction using the inferred spikes is more accurate and suggests that there is a mismatch

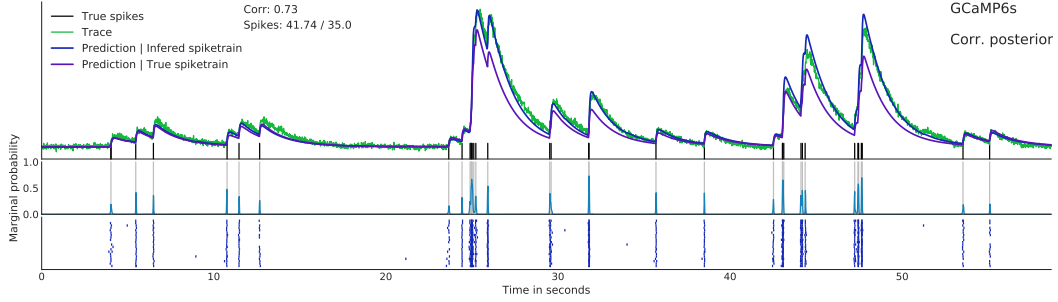

Figure 3: **Inference and reconstruction using the DS-NF algorithm on GECI data**. The reconstruction based on the inferred spike trains (blue) shows that the algorithm converges to a good joint model while the reconstruction based on the true spikes (purple) shows a mismatch of the generative model for high activity which results in an overestimate of the overall firing rate.

between the MLphys model and the true data-generating generating process. Developing more accurate generative models would therefore likely further increase the performance of the algorithm.

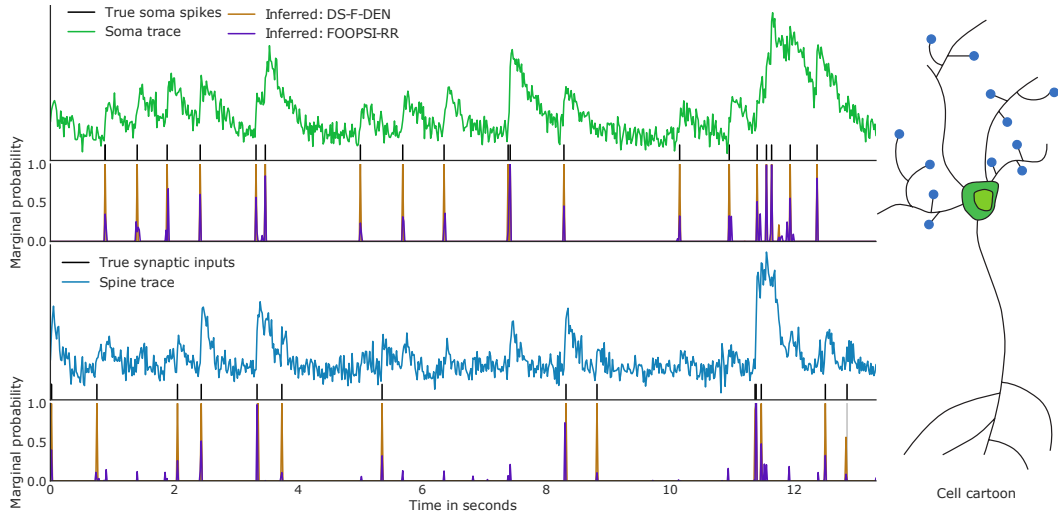

Figure 4: **Inference of somatic spikes and synaptic input spikes from simulated dendritic imaging data.** We simulated imaging data from our generative model, and compared our approach (DS-F-DEN) to an analysis inspired by [2] (Foopsi-RR), and found that our method can extract synaptic inputs more accurately. Traces at the soma and spines are used to infer somatic spikes and synaptic inputs at spines. Top: somatic trace and predictions. DS-F-DEN produces better predictions at the soma since it uses all traces to infer global events. Bottom: spine trace and predictions. DS-F-DEN performs better in terms of extracting synaptic inputs.

### 3.4 Extracting putative synaptic inputs from calcium imaging in dendritic spines

We generalized the DeepSpike variational-inference approach to perform simultaneous inference of backpropagating APs and synaptic inputs, imaged jointly across the entire neuronal dendritic arbor. We illustrate this idea on synthetic data based on the DS-F-DEN generative model (5). We simulated 15 cells each with 10 dendritic spines with a range of firing rates and noise levels. We then used a multi-input multi-output convolutional neural network (CNN, DS-F) in the non-amortized setting to infer a fully-factorized Bernoulli posterior distribution over global action potentials and local synaptic events.

We compared our results to an analysis technique inspired by [2] which we call Foopsi-RR. We first apply constrained deconvolution [33] to somatic and dendritic calcium traces, and then use robust

linear regression to identify and subtract deconvolved components of the spine signal that correlated with global back-propagated action potential. Compared to the method suggested by [2], our model is significantly more accurate. The average correlation of our model is 0.84 for soma and 0.78 for spines, whereas for Foopsi-RR the average correlation is 0.66 for soma and 0.60 for spines (Table 1).

## 4 Discussion

Spike inference is an important step in the analysis of fluorescence imaging. We here propose a strategy based on variational autoencoders that combines the advantages of generative [7] and discriminative approaches [15]. The generative model makes it possible to incorporate knowledge about underlying mechanisms and thus learn from unlabeled data. A simultaneously-learned recognition network allows fast test-time performance, without the need for expensive optimization or MCMC sampling. This opens up the possibility of scaling up spike inference to very large neural populations [34], and to real-time and closed-loop applications. Furthermore, our approach is able to estimate full posteriors rather than just marginal firing rates.

It is likely that improvements in performance and interpretability will result from the design of better, biophysically accurate and possibly dye-, cell-type- and modality-specific models of the fluorescence measurement process, the dynamics of neurons [28] and indicators, as well as from taking spatial information into account. Our goal here is not to design such models or to improve accuracy *per se*, but rather to develop an inference strategy which can be applied to a large class of such potential generative models without model-specific modifications: A trained recognition model that can invert, and provide fast test-time performance, for any such model while preserving performance in spike-detection.

Our recognition model is designed to serve as the common approximate posterior for multiple, possibly heterogeneous populations of cells, requiring an expressive model. These assumptions are supported by prior work [15] and our results on simulated and publicly available data, but might be suboptimal or not appropriate in other contexts, or for other performance measures. In particular, we emphasize that our comparisons are based on a specific data-set and performance measure which is commonly used for comparing spike-inference algorithms, but which can in itself not provide conclusive evidence for performance in other settings and measures. Our approach includes rich posterior approximations [35] based on RNNs to make predictions using longer context-windows and modelling posterior correlations. Possible extensions include causal recurrent recognition models for real-time spike inference, which would require combining them with fast algorithms for detecting regions of interest from imaging-movies [10, 36]. Another promising avenue is extending our variational inference approach so it can also learn from available labeled data to obtain a semi-supervised algorithm [37].

As a statistical problem, spike inference has many similarities with other analysis problems in biological imaging– an underlying, sparse signal needs to be reconstructed from spatio-temporal imaging observations, and one has substantial prior knowledge about the image-formation process which can be encapsulated in generative models. As a concrete example of generalization, we proposed an extension to multi-dimensional inference of inputs from dendritic imaging data, and illustrated it on simulated data. We expect the approach pursued here to also be applicable in other inference tasks, such as the localization of particles from fluorescence microscopy [38].

## 5 Acknowledgements

We thank T. W. Chen, K. Svoboda and the GENIE project at Janelia Research Campus for sharing their published GCaMP6 data, available at http://crcns.org. We also thank T. Deneux for sharing his results for comparison and comments on the manuscript and D. Greenberg, L. Paninski and A. Mnih for discussions. This work was supported by SFB 1089 of the German Research Foundation (DFG) to J. H. Macke. A. Speiser was funded by an IMPRS for Brain & Behavior scholarship by the Max Planck Society.

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
