[Reviews · NeurIPS 2017]

Reviewer 1



This paper presents a variational auto-encoder-style method for extracting spike-trains from two-photon fluorescence microscopy time traces. This is a task that scientists are interested in for the purposes of interpreting one- two- or three- photon microscopy data, which are becoming a staple method in neuroscience. The authors introduce 3 different spike-to-fluorescence models and use one of two neural networks to create an decoder to recover the (discritized) spike trains from the fluorescence. Overall, the method does show some promise. That said, my biggest concern with this work lies with the characterization of the method. Deep-networks just like any other algorithm, need to have an understanding of their trade-offs and limitations. It is especially important in an applications that will be used to inform scientific conclusion to understand aspects of the method such as the the bias and sensitivity to model mis-match. Unfortunately, I feel that the results section fall short of such a characterization and additional simulations or theory needs to be established to instill confidence in this method. I have detailed below specific comments that I feel the authors should address to improve the manuscript. Comments: 1) Is this problem even solvable in general? In the linear case (the first, most basic model) it seems that spikes can be inferred, however the nonlinearities used in the following models can make spikes indistinguishable without adequate time between spikes. Is there a limit to the inter-spike times that this method can distinguish in the non-linear case? Is this a reasonable limit for neural data? 2) In a related note, all the data (and simulations) in this work use 60Hz sampling rates for the fluorescence traces. Much of the fluorescence calcium literature (especially 2-photon imaging, which seems to be becoming dominant) sampling rates as low as 10-15Hz are not unheard of. How does this method perform when the sampling rates fall to 30Hz or 10Hz, when spikes may truly be indistinguishable? Additionally, to have the discrete spike-train output retain a sampling rate on the order of a refractory period, would the proposed networks have an a-symmetry between the input and output rates, or would the network need to be changed to allow for multiple spikes per bins? 3) On line 66, the authors claim to use a "powerful recurrent neural network" model. I assume the authors don't mean 'powerful' in the statistical testing sense, so such embellishment is unnecessary. 4) The authors often claim that the network learned in one sense can generalize well to other settings (e.g. line 61). All the testing and training were done with i.i.d. spikes constant rates chosen from one of three values. I did not see in the results any testing from the network trained with this constrained data-set on fluorescence traces generated from more complex spike-rates. Do the authors have evidence of this generalization, or is this a statement provided based on the use of VAEs in other contexts? If it is the former, the authors should provide these results to validate their claim. If it is the latter, the authors should verify that this generalization holds true for their application, as properties of VAEs do not necessarily translate across domains. 5) The authors present three different models with various levels of complexity, however I do not see any results as to how the proposed network inference methodology changes with respect to each of these networks. It would be informative to see how the results, say on a single V1 trace, change with respect to the model used to train the network. 6) Related to the previous point, sometimes it is unclear which model is being used in which networks for the various results (e.g. what model is used for the V1 results?). 7) In Figure 3, the reconstruction base on the true spike train using (was this using the SCF or SCDF model?) is not as accurate as the results of the spikes inferred using the network trained on that model. How is this possible? This discrepancy warrants at least some discussion. Clearly there is a model-mismatch of some kind. Is the network's ability to achieve better prediction mean that the method has a built-in bias in the estimated spikes that stems from the model mismatch? This seems like an important point that is glossed over. A systematic bias in the inferred spikes can create errors in scientific analyses applied to the resulting spike trains. 8) The work "amortize" is useless in the context of machine learning and only serves to further obfuscate our field further with respect to outside audiences that may seek to use these methods. I very highly recommend omitting this word from the paper. 2) Acronyms should be defined where first used (e.g. in all the sub-sub-section titles for the various models) 3) Most of the figures could do with bigger font sizes

Reviewer 2



The paper described a variational autoencoder (VAE) framework for spike time inference from calcium imaging data. The authors propose 2 recognition models, one with a recurrent network component that accounts for temporal correlations in spike times, and another that is purely feed-forward but that nonetheless accounts temporal correlations in the calcium signal by convolving over a window. The authors also use a principled model for the generative model of the calcium imaging data, conditioned on spike times. The premise of the paper is summarized in lines 60-61, “Once trained to perform spike inference on one dataset, the recognition model can efficiently generalize to other similar datasets without retraining.” The written quality of the paper is mostly exceptional with complex ideas being described very fluidly with the appropriate degree of background and appropriate references to relevant literature. The idea is appropriately novel and the authors are using state of the art inference and modeling approaches. I would have liked to score this paper higher as I appreciate the idea, the written quality, and the technical quality of the paper. However, there is at least one very troubling element to this paper. Specifically, it is not clear precisely what the authors are proposing practitioners should do with this method. Should a scientist go out and find some instantiation of this model and use it directly on their data without retraining for their specific setup? Should they not retrain their network for each new dataset, assuming that each new dataset will have a different class of cell being imaged and a different version of GcAMP? Wouldn’t the model have to be retrained when the training set has a different dynamic range of firing rates than the test set? It is difficult to believe, and the authors have not demonstrated, that the method is transferrable across datasets, only that it has good predictive performance on a holdout set. This is not to say that the method is worthless. I see clear value in the method. I only think that the “amortized” part of this method is unsubstantiated. I also have two complaints regarding specific points in the paper. For one, the authors propose four different generative models (equations (2)-(5)) but appear only to have used one of them in their simulation and real data experiments. Also, Figure 3 indicates that the prediction from the inferred spike train is considerably better than the prediction from the true spike train, especially at high spike rates, suggesting some kind of bias in the spike inference. Could the authors address this?

Reviewer 3



This paper describes an innovative DNN-based approach for inferring spikes from calcium imaging measurements. This is an extremely important problem, as calcium imaging has rapidly become one of the standard tools for assessing activity in large neural populations. While a number of approaches exist to solve the problem, mostly based on generative models, the problem is far from solved (see e.g. recent quantitative evaluation by ref. 15 in the paper). The authors propose a creative solution based on a combination of a generative model and a recognition model based on convolutional and recurrent neural network architectures. Conceptually, this is a very strong contribution and their quantitative evaluation indicates that their approach achieves state-of-the-art performance on simulated data and a small set of actual recordings. The authors also outline interesting future possibilities for separating backpropagating spikes and dendritic signals from one another using the model. I only have a few minor comments: The authors understate the amount of data available on crcns.org – by now there are additional datasets. I think it is fine for this paper to focus on one of them, the relatively straightforward case of recordings from individual neurons at large magnification. How performance transfers to the case of population recordings is a different question. The authors should briefly note that somewhere. I am not sure I understand the setup of the simulations described in lines 2017ff – if you train your model on 50 cells from one model and evaluate it one 30 others simulated from the same model, how is it surprising that performance transfers? Rather, why can it even be a little lower? Is there variability in the model parameters between the cells? I feel like I am missing something here. Table 1 should note that values are correlation coefficients. Figure 4 could be cleaned up – e.g. for placement of the legends and maybe avoiding bar plots in B.